# Assessment of Immunogenicity and Efficacy of CV0501 mRNA-Based Omicron COVID-19 Vaccination in Small Animal Models

**DOI:** 10.3390/vaccines11020318

**Published:** 2023-01-31

**Authors:** Nicole Roth, Janina Gergen, Kristina Kovacikova, Stefan O. Mueller, Lorenz Ulrich, Jacob Schön, Nico Joel Halwe, Charlie Fricke, Björn Corleis, Anca Dorhoi, Donata Hoffmann, Martin Beer, Domenico Maione, Benjamin Petsch, Susanne Rauch

**Affiliations:** 1CureVac SE, 72076 Tübingen, Germany; 2Institute of Diagnostic Virology, Friedrich Loeffler Institut, 17493 Griefswald-Insel Riems, Germany; 3Institute of Immunology, Friedrich-Loeffler-Institut, 17493 Greifswald-Insel Riems, Germany; 4GSK, 53100 Siena, Italy

**Keywords:** COVID-19 variant, SARS-CoV-2, mRNA vaccine, virus neutralizing antibody titer

## Abstract

Severe acute respiratory syndrome coronavirus-2 (SARS-CoV-2) Omicron and its subvariants (BA.2, BA.4, BA.5) represented the most commonly circulating variants of concern (VOC) in the coronavirus disease 2019 (COVID-19) pandemic in 2022. Despite high vaccination rates with approved SARS-CoV-2 vaccines encoding the ancestral spike (S) protein, these Omicron subvariants have collectively resulted in increased viral transmission and disease incidence. This necessitates the development and characterization of vaccines incorporating later emerging S proteins to enhance protection against VOC. In this context, bivalent vaccine formulations may induce broad protection against VOC and potential future SARS-CoV-2 variants. Here, we report preclinical data for a lipid nanoparticle (LNP)-formulated RNActive^®^ N1-methylpseudouridine (N1mΨ) modified mRNA vaccine (CV0501) based on our second-generation SARS-CoV-2 vaccine CV2CoV, encoding the S protein of Omicron BA.1. The immunogenicity of CV0501, alone or in combination with a corresponding vaccine encoding the ancestral S protein (ancestral N1mΨ), was first measured in dose-response and booster immunization studies performed in Wistar rats. Both monovalent CV0501 and bivalent CV0501/ancestral N1mΨ immunization induced robust neutralizing antibody titers against the BA.1, BA.2 and BA.5 Omicron subvariants, in addition to other SARS-CoV-2 variants in a booster immunization study. The protective efficacy of monovalent CV0501 against live SARS-CoV-2 BA.2 infection was then assessed in hamsters. Monovalent CV0501 significantly reduced SARS-CoV-2 BA.2 viral loads in the airways, demonstrating protection induced by CV0501 vaccination. CV0501 has now advanced into human Phase 1 clinical trials (ClinicalTrials.gov Identifier: NCT05477186).

## 1. Introduction

Since detection in humans in December 2019, severe acute respiratory syndrome coronavirus 2 (SARS-CoV-2) has spread globally to cause the coronavirus disease 2019 (COVID-19) pandemic. As of December 2022, it is estimated that COVID-19 has caused more than 6.6 million deaths worldwide [1].

In this time, a multitude of SARS-CoV-2 variants have evolved, with some classed as variants of concern (VOC). The Omicron BA.1 variant, first detected in Botswana and South Africa in November 2021, has rapidly spread worldwide and is characterized by a high mutational burden compared with other VOCs. The Omicron BA.5 subvariant has quickly increased in global prevalence since it was first reported in February 2022 [2]. After being declared a VOC by the WHO in May 2022, the US FDA recommended that the Omicron BA.4/5 spike protein should be added to the current vaccine composition to create an updated two-component (bivalent) booster vaccine [3]. New Omicron subvariants have continued to emerge, including, BQ.1, a sub lineage of BA.5, and XBB, a recombinant of BA.2.10.1 and BA.2.75, both of which have altered antibody evasion properties [4]. Rapid antigenic drift in Omicron and its subvariants has led to increased transmissibility and evasion of humoral responses elicited by vaccines based on ancestral S protein, leading to pressing concerns around vaccine resistance and efficacy [5,6,7,8].

Despite the successful implementation of mass COVID-19 vaccination programs across the globe, Omicron and its subvariants continue to cause significant morbidity and mortality. As infection and reinfection in both vaccinated and unvaccinated individuals occurs increasingly with SARS-CoV-2 variants, and reinfection becoming associated with more severe acute and post-acute sequelae compared with a primary infection, SARS-CoV-2 continues to have significant global economic consequences [9,10,11,12,13].

BNT162b2 (Comirnaty, Pfizer/BioNTech) and mRNA-1273 (Spikevax, Moderna) monovalent mRNA COVID-19 vaccines, approved in December 2020 and April 2021 respectively, were generally well tolerated and highly effective at reducing the severity of COVID-19 symptoms caused by the ancestral strain [14,15]. However, the emergence of SARS-CoV-2 variants has highlighted the necessity of long-term protection and the need for booster doses, and the importance of developing SARS-CoV-2 variant-matched vaccines [16]. Bivalent mRNA vaccine boosters that include ancestral and either BA.1 or BA.4/5 components have recently been authorized in Europe, Australia and the United States.

The emergence of Omicron BA.1 prompted the development of CV0501, an RNActive^®^ vaccine based on optimizations introduced in our second-generation SARS-CoV-2 vaccine CV2CoV [17,18,19]. CV0501 is a monovalent mRNA vaccine candidate using modified nucleosides, encoding the S protein of Omicron BA.1 and maintaining the same pre-fusion stabilizing mutations in S protein as previously described [17,20]. mRNA vaccines encoding for ancestral S protein (ancestral N1mΨ) and a different BA.1 isolate (BA.1 nonclinical) were employed as comparators.

CV0501 immunogenicity as a primary vaccination was evaluated in a dose response study in Wistar rats and compared with immunization with BA.1 nonclinical and ancestral N1mΨ vaccines. A further study in rats primed with ancestral N1mΨ evaluated neutralizing antibody (nAb) responses induced by boosting with monovalent CV0501, ancestral N1mΨ or a bivalent CV0501/ancestral N1mΨ vaccine. Immunogenicity and protective efficacy of CV0501 were evaluated in an Omicron BA.2 challenge model in Syrian hamsters.

## 2. Methods and Materials

### 2.1. Vaccines

All vaccines tested were N1-methylpseudouridine (N1mΨ) modified, mRNA-based vaccines that are formulated in identical lipid nanoparticles (LNP). mRNAs contain a cleanCap followed by the 5′ UTR from the human hydroxysteroid 17-beta dehydrogenase 4 gene (HSD17B4) and a 3′ UTR from the human proteasome 20S subunit beta 3 gene (PSMB3), followed by a histone stem–loop and a poly(A)100 stretch. The constructs are formulated using LNP technology from Acuitas Therapeutics, composed of ionizable amino lipid, phospholipid and cholesterol and PEGylated lipid.

All mRNAs encode for full length SARS-CoV-2 spike (S) protein containing stabilizing K986P and V987P mutations [20,21]. The S proteins encoded in the vaccines are either derived from ancestral (EPI_ISL_402124) or BA.1 SARS-CoV-2 variants. The initial Omicron BA.1 isolates featured had 15 mutations in their receptor binding domain (RBD), but three of these, (K417N, N440K and G446S) are not conserved in all later isolates of this subvariant. Thus, the clinical candidate, CV0501, does not have these three mutations (EPI_ISL_6699769), while BA.1 nonclinical has the three mutations, but is otherwise identical to CV0501 (EPI_ISL_6640916). The immunological characteristics of these two variants were compared to investigate the impact of the three mutations. All specific mutations are listed in Table 1.

For the generation of the bivalent CV0501/ancestral N1mΨ vaccine, both vaccine components were mixed extemporaneously in a 1:1 ratio prior to injection.

### 2.2. Animal Models

Female Wistar rats, aged 7–8 weeks, provided and handled by Preclinics (Potsdam, Germany) were used for dose response and booster studies. Male Syrian hamsters aged 11 weeks purchased from Janvier Labs (Le Genest-Saint-Isle, France), housed at the Friedrich-Loeffler-Institut, were used for the challenge studies. All procedures using SARS-CoV were carried out in approved biosafety level 3 facilities.

The animal studies were conducted in accordance with German laws and guidelines for animal welfare. To comply with the principles of the 3Rs, each animal experiment was performed once. The protocol for the rat studies received the appropriate local and national ethics committees’ approvals (2347-5-2021 LAVG Brandenburg) and the protocol for the hamster studies was approved by the ethics committee of the State Office of Agriculture, Food safety, and Fishery in Mecklenburg, Western Pomerania (LALLF M-V: 7221.3-1-036/21).

### 2.3. Dose Response and Booster Studies

For the dose response studies, rats (*n* = 8/group) were anaesthetized with isoflurane and injected intramuscularly (i.m.) on Days 0 and 21 with 100 µL volume of 2, 8 or 20 µg of CV0501 or BA.1 nonclinical, or 100 µL volume of 8 µg ancestral N1mΨ vaccines diluted with 0.9% NaCl. Control rats received 100 µL of 0.9% NaCl alone (*n* = 6). On Days 14, 21 and 42 post immunization, blood was collected into Z-clot activator tubes (Sarstedt; Nümbrecht, Germany) and incubated at room temperature for 30 min to allow for coagulation. Tubes were then centrifuged, serum removed and stored at <−70 °C. Serum samples were shipped on dry ice from Preclinics to VisMederi for analyses.

For the booster studies, rats were anaesthetized with isoflurane and injected i.m. on Days 0 and 21 with 100 µL of 2 or 8 µg of ancestral N1mΨ or BA.1 nonclinical (*n* = 8). Control rats received 100 µL of 0.9% NaCl alone (*n* = 6). On Days 105 and 189, rats were given booster doses (third and fourth doses, 100 µL volume) of 2 or 8 µg of ancestral N1mΨ, CV0501 or bivalent CV0501/ancestral N1mΨ ([1 µg/1 µg] or [4 µg/4 µg]). On Days 21, 42, 77, 105, 133, 161, 189 and 217, blood was collected for serum antibody analyses, as described above. For further details see Appendix A.

### 2.4. Neutralizing Antibody Titers

Sera were heat inactivated at 56 °C for 30 min. Serial dilutions of serum were incubated for 60 min at 37 °C, with 100 TCID_50_ (median tissue culture infectious dose) of SARS-CoV-2. The virus strains used to assess nAb titers included ancestral, Beta, Delta, BA.1 (with 15 RBD mutations), BA.2 and BA.5.

Infectious virus was quantified by incubating 100 µL of virus-serum mixture with a confluent layer of 6+ followed by incubation for 3 days (with ancestral SARS-CoV-2) or 4 days (SARS-CoV-2 Beta, Delta, BA.1. BA.2 or BA.5) at 37 °C and microscopical scoring for cytopathogenic effect (CPE). A back-titration was performed for each run to verify the range of TCID_50_ in the working virus stock.

Neutralizing antibody titers were calculated according to the method described by Reed & Muench [22]. If no neutralization was observed (microneutralization test < 10), an arbitrary value of 5 was attributed. Analyses were carried out by VisMederi (Siena, Italy).

### 2.5. Enzyme-Linked Immunosorbent Spot (ELISpot) Assay

Splenocytes from rats were isolated and single-cell suspensions were prepared in supplemented medium. A total of 5 × 10^5^ splenocytes per well (200 µL volume) were stimulated for 24 h at 37 °C using a SARS-CoV-2 peptide library (JPT, PM-SARS2-SMUT08-1, Berlin, Germany) at 1 µg/mL. T cells were analyzed using ELISpot according to the manufacturer’s protocol ELISpot Rat interferon-gamma (IFNγ), Cat: EL585 by R&D Systems, Minneapolis, MN, USA).

### 2.6. ACE2 Binding Inhibition Assay Methodology

The ACE2 binding inhibition assay was based on a previously published multiplex competitive binding assay [23]. Briefly, different RBD antigens from SARS-CoV-2 variants were immobilized on spectrally distinct populations of magnetic MagPlex beads (Luminex, Austin, TX, USA) by Anteo coupling (AMG Activation Kit for Multiplex Microspheres, Anteo Technologies, Brisbane, QLD, Australia (#A-LMPAKMM-400)), as previously described [23]. Coupled beads were stored at 4 °C and then combined into a 25× bead mix. The antigens used in these experiments are listed in Appendix A.

25 µL of diluted rat sera were then mixed 1:1 with 1× bead mix (25× bead mix diluted in assay buffer containing biotinylated ACE2), generating a final dilution of either 1:1600 or 1:3200, in a 96-half-well plate and incubated for 2 h on a thermomixer (750 rpm, 20 °C). Following this, samples were washed to remove unbound ACE2 using an automated magnetic plate washer. To detect bound ACE2, 2 µg/mL Streptavidin-Phycoerythrin (PE) (MOSS, Cat# SAPE-001, Dunn Labortechnik GmbH, Asbach, Germany) was added and the plate incubated for a further 45 min. After washing, beads were resuspended in 100 µL of wash buffer and mixed for 3 min at 1000 rpm [24]. Plates were read on a FLEXMAP3D instrument (Luminex, Austin, TX, USA) using the following settings: 80 µL (no timeout), 50 events, Gate 7500–15,000, Standard PMT. Each plate included 3 wells with 150 ng/mL ACE2, 2 blank wells and 3 wells with a quality control (QC) sample as controls [25]. For each variant, ACE2 inhibition was calculated as a percentage, with 100% indicating maximum ACE2 inhibition and 0% indicating no ACE2 inhibition.

### 2.7. Hamster Challenge Model

#### 2.7.1. Immunizations

Male Syrian hamsters received 50 µL of either 8 or 24 µg of CV0501 or 0.9% NaCl (sham controls) (*n* = 9/group) via i.m. injection using BD Micro-Fine 0.5 mL insulin syringes with 30 G × 8 mm needle, in the left outer thigh (biceps femoris) on Day 0. Second doses were administered in the right outer thigh (biceps femoris) at Day 28.

#### 2.7.2. Challenge Infection

In week 8, all hamsters were challenged with 1 × 10^5^ TCID_50_ SARS-CoV-2 Omicron BA.2 (B. Haagmans, Rotterdam, The Netherlands). Challenge virus was administered as 0.1 mL intranasally (i.n.) (0.05 mL in each nostril) and dose was confirmed via virus back-titration. Animals were weighed daily until sacrifice. Six animals per group were sacrificed on Day 4 post-challenge for determination of viral titers in the conchae, trachea and lungs. The remaining animals (3 per group) were sacrificed on Day 14 post-challenge.

#### 2.7.3. Viral RNA Detection and Quantification

Approximately 0.1 cm^3^ samples of each organ were homogenized in a 1 mL mixture composed of equal volumes of Minimum Essential Medium (MEM) Hank’s balanced salts and MEM Earle’s balanced salts (containing 2 mM L-glutamine, 850 mg L^−1^ NaHCO_3_, 120 mg L^−1^ sodium pyruvate and 1% penicillin–streptomycin) at 300 Hz for 2 min using a TissueLyser II (Qiagen, Hilden, Germany).

Nasal washings from hamsters (*n* = 9/group) were obtained on Day 2 and 4 post-challenge by administering 0.2 mL volume of phosphate-buffered saline directly into each nostril and subsequently collecting the efflux.

Nucleic acid was extracted from 100 µL of nasal washes after a short centrifugation step or 100 µL of organ sample supernatant using the NucleoMag Vet kit (Macherey Nagel, Dueren, Germany). Each extracted sample was eluted in 100 µL. Viral RNA genome was detected and quantified by reverse transcriptase quantitative polymerase chain reaction (RT-qPCR) on a BioRad real-time CFX96 detection system (BioRad, Hercules, United States). The target sequence for amplification was the viral RNA-dependent RNA polymerase [26].

Genome copies of RNA per µL were calculated based on a quantified standard RNA, where absolute quantification was carried out by the QX200 Droplet Digital PCR System in combination with the 1-Step RT-ddPCR Advanced Kit for Probes (BioRad, Hercules, CA, USA). The limit of detection was calculated as 10 copies per reaction.

#### 2.7.4. Surrogate ELISA

This surrogate ELISA measures the percentage inhibition of BA.1 binding. Sera were screened at a 1:10 dilution using a competitive enzyme linked immunosorbent assay with the S-RBD horseradish peroxidase (HRP) for Omicron BA.1 (SARS-CoV-2 sVNT L00847-A and S-RBD HRP Z03730, GenScript, Rijswijk, The Netherlands) according to the manufacturer’s instructions. A reduction in optical density (OD) of ≥30% compared with the mean OD of the negative control was considered positive for seroconversion of antibodies against BA.1.

#### 2.7.5. Neutralizing Antibody Titers

Neutralizing antibody titers were determined at the Friedrich-Loeffler-Institut. Briefly, sera were pre-diluted ≤1/64 with Dulbecco’s modified Eagle’s medium (DMEM) in a 96-well deep well master plate. Subsequently, 100 TCID_50_/well of the respective SARS-CoV-2 (Omicron BA.1 or Omicron BA.2) virus dilution was added and incubated for 1 h at 37 °C. Lastly, 100 µL of trypsinated Vero E6 cells (cells of 1 confluent TC175 flask per 100 mL) in DMEM with 1% penicillin/streptomycin supplementation was added to each well. After 72 h of incubation at 37 °C, the cells were evaluated by light microscopy for a specific CPE. The nAb titer was the dilution with no visible CPE and the viral titers were confirmed by back-titrations.

#### 2.7.6. Statistical Analysis

Statistical analyses were performed using GraphPad Prism (version 9.0) software. Analysis of variance (ANOVA) was performed, followed by non-parametric Kruskal-Wallis post-tests or Dunnett‘s multiple comparison tests (vaccinated against sham) for the body weight analysis of hamsters after challenge infection. *p*-values were designated as: * *p* ≤ 0.05, ** *p* ≤ 0.01, *** *p* ≤ 0.001 and *** *p* ≤ 0.0001.

## 3. Results

### 3.1. CV0501 Induces nAb Responses against SARS-CoV-2 Variants in Wistar Rats

Virus neutralizing antibodies (nAbs) against BA.1 were measured in sera from rats immunized with two doses of ancestral N1mΨ (8 µg), CV0501 (2, 8 or 20 µg) or BA.1 nonclinical (2, 8 or 20 µg) vaccines (Figure 1). Compared with NaCl-treated controls, the first dose (Day 0) of CV0501 and BA.1 nonclinical induced robust nAb responses against the BA.1 variant (measured on Days 14 and 21), with titers increasing between 8 and 19-fold between Day 21 and Day 42. Compared with the 8 µg ancestral N1mΨ vaccine, BA.1 nonclinical (8 and 20 µg) induced significantly higher BA.1-specific nAb titers at all timepoints. Similar differences were observed for CV0501. No significant differences were detectable between CV0501 and BA.1 nonclinical at any time point.

At Day 14 and 21, following a single dose of CV0501 (8 µg), 12-fold and 3.4-fold higher nAb titers against BA.1, respectively, were noted compared with serum from rats receiving a single dose of 8 µg ancestral N1mΨ vaccine. At Day 42, serum from rats following the second immunization of CV0501 (8 µg) had 2.3-fold higher nAb titers against BA.1 compared with rats receiving a second immunization of 8 µg ancestral N1mΨ vaccine (Figure 1).

Next, the induction of cross-nAbs following two immunizations of ancestral N1mΨ vaccine, CV0501 or BA.1 nonclinical, were compared on Day 42 (Figure 2). Analysis of rats vaccinated with CV0501 and BA.1 nonclinical demonstrated a dose-dependent increase in cross-nAbs between 2 µg and 8 µg groups across all tested SARS-CoV-2 variants. Induction of cross-nAbs was comparable for the 8 µg and 20 µg groups of CV0501 against all variants, likely due to a saturation effect at high doses. CV0501 and BA.1 nonclinical induced overall comparable nAb titers against ancestral, Beta, Delta, BA.2 and BA.5 variants. However, a trend towards increased nAb titers following vaccination with CV0501 compared with BA.1 nonclinical was observed for all variants and this was statistically significant for BA.2 neutralization in the 8µg groups (Figure 2d). An exception to this was Beta SARS-CoV-2 neutralization, whereby immunization with BA.1 nonclinical elicited numerical higher nAb titers compared with CV0501 (Figure 2b).

Immunization with 8 µg ancestral N1mΨ yielded significantly higher levels of cross-nAbs against ancestral, Delta and BA.5 SARS-CoV-2 compared with 8 µg BA.1 nonclinical (Figure 2a,c,e). No significant differences in cross-neutralization of Beta or BA.2 were detectable between ancestral N1mΨ and BA.1 nonclinical in 8 µg groups (Figure 2b,d). Furthermore, no significant differences in cross-neutralization of all SARS-CoV-2 variants were detectable between ancestral N1mΨ and CV0501 in the 8 µg groups, except for Beta neutralization which was significantly increased upon vaccination with ancestral N1mΨ (Figure 2b). In a side-by-side comparison of different SARS-CoV-2 variants, two doses of CV0501 (2, 8 or 20 µg) induced highest nAb titers against Omicron BA.1, BA.2 and BA.5 compared with ancestral, Beta and Delta neutralization in rats (Appendix A).

Spike-specific IFN-γ-producing T cells stimulated with an Omicron peptide pool were analysed from rat splenocytes isolated on Day 42 via ELISpot (Appendix A). Comparison of the 8 µg doses of each vaccine showed that CV0501 induced significantly higher numbers of IFN-g spots compared with the ancestral N1mΨ vaccine, while no significant differences were observed between ancestral N1mΨ and BA.1 nonclinical or CV0501 and BA.1 nonclinical at 8 µg.

### 3.2. CV0501 Boosting Induces Cross-nAbs against SARS-CoV-2 Variants in Ancestral-Primed Wistar Rats

Viral nAb titers against SARS-CoV-2 BA.1 were assessed in rats primed with 2 doses of ancestral N1mΨ or BA.1 nonclinical (2 µg or 8 µg; Days 0 and 21) and boosted with 2 µg or 8 µg of either CV0501, ancestral N1mΨ or a bivalent combination of ancestral N1mΨ and CV0501 (CV0501/ancestral N1mΨ) on Day 105 and Day 189 (Figure 3 and Appendix A, Table 2).

The lowest nAb levels against BA.1 were observed in ancestral N1mΨ-primed/ancestral N1mΨ boosted animals and the highest nAb levels against BA.1 were observed in BA.1 nonclinical primed/CV0501 boosted animals (Table 2, Figure 3c,d).

BA.1-specific nAb titers increased between Day 21 and Day 42, following the second priming dose in all groups. In general, titers induced by priming were numerically higher in the 8 µg groups compared with the two µg groups. The highest nAb titers against BA.1 upon priming were observed in the serum of BA.1 nonclinical-primed rats (Figure 3d, Table 2). Following a third vaccine dose (first boost) on Day 105, increases in nAb titers against BA.1 were observed for all vaccine combinations (Day 105 vs. Day 133), although the effect was more pronounced in the 2-µg ancestral N1mΨ-primed groups (Figure 3a,b). The highest-fold increase in nAbs between Day 105 and Day 133 in the 2 µg groups primed with ancestral N1mΨ was observed upon CV0501 boosting (5.6-fold) (Figure 3a), followed by bivalent CV0501/ancestral N1mΨ boosting (1.9-fold) (Figure 3b) and ancestral N1mΨ boosting (1.5-fold) (Figure 3c). Priming and boosting with BA.1 encoding vaccines induced the lowest increase of 1.2 fold between Day 105 and Day 133 (Table 2) (Figure 3d). The differences were less pronounced in the 8 µg dose groups although the BA.1 nonclinical-primed/CV0501 boost group showed a 2.3-fold increase in nAbs between Day 105 and Day 133 (Table 2). In all 2 µg and 8 µg vaccine dose groups, the second booster dose (Day 189) increased BA.1-specific nAb titers (Day 189 vs. Day 217), although none of the increases were statistically significant (Figure 3). The fold increase in neutralizing titers after second boost (Day 189) followed the trend of the first boost (Day 105) (Table 2).

On Day 217 (28 days after the fourth vaccine/second booster dose), the neutralizing activity of rat serum was compared against the ancestral, Delta and Omicron BA.1, BA.2 and BA.5 SARS-CoV-2 variants (Figure 4). All vaccine combinations induced high levels of nAbs against all of the SARS-CoV-2 variants tested, with each 8 µg vaccine regimen outperforming its equivalent 2 ug counterpart in all cases (Table 3).

All regimens (2 µg and 8 µg) featuring ancestral N1mΨ priming induced higher nAb titers against the ancestral and Delta variants than the regimen combining BA.1-based priming and boosting (Figure 4a,b, Table 3) with statistically significant differences observed for CV0501/ancestral N1mΨ and ancestral N1mΨ (2 and 8 µg doses) but not the 8 µg dose for ancestral N1mΨ against Delta. On the other hand, the regimen employing only BA.1 based vaccines produced numerical higher nAb titers against the BA.1 and BA.2 variants than the regimens based on ancestral N1mΨ priming (Figure 4c,d, Table 3). Similar levels of nAbs against BA.5 were induced in all the 2 µg groups. All 8 µg groups, primed with ancestral N1mΨ, followed by booster vaccination with CV0501 as a monovalent or bivalent, had significantly increased BA.5 nAb titers compared with ancestral N1mΨ boosting (Figure 4e).

Boosting with bivalent CV0501/ancestral N1mΨ and monovalent CV0501 induced nAb titers against all variants tested. There was a trend towards increased responses against ancestral, Delta, BA.2 and BA.5 in the 2 µg groups of animals boosted with bivalent CV0501/ancestral N1mΨ compared with monovalent CV0501 vaccine that was non-significant for all variants except for against ancestral (Figure 4a,b,d,e).

Priming with ancestral N1mΨ resulted in broader ACE2 binding inhibition against RBDs of a range of SARS-CoV-2 variants, compared with priming with BA.1 nonclinical (Figure 5). Among the 2 µg vaccine regimens, priming with ancestral N1mΨ and boosting with the bivalent CV0501/ancestral N1mΨ vaccine tended to increase ACE2 binding inhibition across the SARS-CoV-2 variants tested compared with all the other combinations assessed (Figure 5a). However, this trend was absent in the 8 µg dose groups, where the inhibition capacity for all the ancestral N1mΨ-primed regimens was comparable (Figure 5b).

### 3.3. CV0501 Protects against Weight Loss and Viral Replication in the Respiratory Tract in Syrian Hamsters

To gain insights into the protective efficacy of CV0501, Syrian hamsters vaccinated twice with 8 µg or 24 µg of CV0501 were challenged with 10^5^ TCID_50_ of Omicron BA.2 four weeks post second vaccination. A buffer vaccinated group challenged in parallel served as sham-control.

Animals were weighed daily during the challenge phase from Days 56 to 70 (Days 0 to 14 post-challenge: Figure 6a). The animals in the sham-immunized group had the highest mean weight loss (6.5%) up to Day 6 post-challenge with weights returning to the baseline level within 8 days, in line with the non-lethal challenge model. At Day 4, animals in the 24 µg CV0501-immunized group had significantly higher relative body weight compared with the sham-immunized. Apart from this, there was no significant difference between the groups throughout the challenge phase.

Viral replication in the upper respiratory tract was evaluated in nasal washes performed on Day 2 and Day 4 post challenge and analyzed via RT-qPCR. For all groups, the BA.2 viral genome load in nasal washes was lower on Day 4 compared with Day 2 (Figure 6b). Levels of viral genomes were readily detectable in all groups. However, vaccination with CV0501 induced a dose-dependent reduction of viral genome copies in the immunized hamsters. Detected reductions were 2 and 9-fold (Day 2) and 4 and 67-fold (Day 4) for the 8µg and 24µg CV0501 vaccinated groups, respectively. Statistically significant lower levels of BA.2 viral genome copies were detected at Day 2 and Day 4 in the nasal washes from animals immunized with 24 µg CV0501, compared to sham immunized controls.

Viral loads were assessed in nasal conchae, trachea and lung samples collected on Day 4 (*n* = 6/group) and Day 14 (*n* = 3/group) post BA.2 challenge (Figure 6c,d). Overall viral genome copies were higher on Day 4 compared with Day 14 tissue samples. In parallel to results in nasal washes, immunization with both 8 µg and 24 µg of CV0501 significantly reduced genome copies in the conchae at both timepoints, compared with sham-vaccinated animals. CV0501 vaccination reduced viral genome copies in the conchae 8- and 45-fold (Day 4) and 4- and 17-fold (Day 14) in 8 µg and 24 µg CV0501 vaccinated groups, respectively. At Day 4 post challenge, lower numbers of viral genome copies were observed in the lungs of both CV0501-immunized groups, compared with sham immunized- group (Figure 6c). Immunization with 8 µg of CV0501 reduced viral genome copies by 3 Log_10_ (cranial), 2.5 Log_10_ (medial) and 3.3 Log_10_ (caudal), compared with sham-immunized hamsters. Immunization with 24 µg of CV0501 elicited reductions of 2.7 Log_10_ (cranial), 3.1 Log_10_ (medial) and 3.8 Log_10_ (caudal), compared with sham controls. On Day 14 post-challenge, BA.2 genome copies were undetectable in the trachea and lungs of any CV0501-immunized animals, although none of the comparisons with sham vaccinated animals were statistically significant in the lower respiratory tract (Figure 6d). CV0501 immunization resulted in a reduction of viral genome copies in the lower respiratory tract, although only reductions observed in the conchae were statistically lower than in the sham vaccinated animals.

The neutralizing ability of sera from CV0501-immunized hamsters against BA.1 (Figure 7a) and BA.2 (Figure 7b) variants was assessed immediately before SARS-CoV-2 challenge (Day 0) and on Days 4 and 14 post-challenge. Prior to challenge (Day 0), CV0501 immunization induced robust dose-dependent nAb titers against BA.1 (Figure 7a) and to a lesser extent against BA.2 (Figure 7b). While no increase in nAbs against either subvariant tested was detected upon challenge in the sham group, BA.2 challenge infection led to an overall increase of nAbs titers against BA.1 and BA.2 in CV0501 vaccinated animals over time. Despite nAb titers against BA.2 being lower on Days 0 and 4 compared with BA.1, the nAb titers against both BA.1 and BA.2 reached similar levels by Day 14. The increase in nAbs was most pronounced in the 8-µg dose group and significant on Day 14 for both BA.1 and BA.2 neutralization compared with sham vaccinated (Figure 7a,b). Due to the low volume of sera available for viral neutralization assays, the starting dilution used for the nAb assay was 1:64. To provide more sensitive nAb data, a surrogate competition ELISA was performed. This ELISA measures the percentage inhibition of BA.1 binding with a cut-off value of 30% being considered as sero-positive (Appendix A). Results showed a similar inhibition after immunization with both 8 µg and 24 µg doses of CV0501, at all timepoints. On Day 0 and 4 post challenge, sham-immunized hamsters demonstrated lower inhibition compared with CV0501. The percentage of inhibition increased in sham-immunized hamsters on Day 14 post challenge, which was not observed in the less sensitive neutralization assay (Figure 7a,b).

## 4. Discussion

Despite the success of global COVID-19 immunization programs resulting in protection against severe COVID-19, widely employed vaccines based on the ancestral S protein provide only short-lived protection against mild to moderate disease and have low vaccine effectiveness against symptomatic disease with Omicron [27,28]. The objective of this study was to assess the immunogenicity and efficacy of vaccine candidate, CV0501, which encodes the S protein of the Omicron variant BA.1, as both a monovalent or bivalent vaccine in combination with S protein of ancestral SARS-CoV-2.

In Wistar rats, CV0501 induced robust, dose-dependent levels of nAbs against homologous SARS-CoV-2 BA.1 upon a single injection and was able to elicit nAb responses against the tested SARS-CoV-2 variants, including Omicron subvariants BA.2 and BA.5, thus showing promise for inducing protection against future Omicron-derived subvariants. In this study, two versions of the BA.1 vaccine, CV0501 and BA.1 nonclinical candidate, were tested. The latter contains three mutations in the receptor-binding domain (RBD), i.e., K417N, N440K and G446S that are not present in the S protein encoded by CV0501. These mutations which were observed in the original BA.1 variant are present in only 60–65% of BA.1 isolates. All three mutations have been reported to be involved in immune evasion from a subset of neutralizing monoclonal antibodies [29,30,31]. However, we observed no significant differences in nAb responses induced by CV0501 and BA.1 nonclinical candidate, which also held true for nAb titers against BA.2 and BA.5 that have the K417N and N440K mutations. Overall, no differences indicative of significant changes in vaccine immunogenicity were detected between the two tested BA.1 vaccines in our experimental system. The sequence of CV0501 was favored since it was more prevalent within the BA.1 subvariant at time of the decision to progress to clinical development.

Our findings from prime/boost studies suggest that CV0501, both as monovalent and bivalent vaccine, is superior to ancestral N1mΨ as a booster vaccine for the induction of BA.1 specific nAbs. Additionally, monovalent and bivalent CV0501 vaccines induced cross-neutralizing antibodies to the ancestral and Delta variants and induced higher responses against Omicron BA.2 and BA.5 variants compared with boosting with ancestral N1mΨ.

As expected, priming and boosting of rats with BA.1 S protein encoding vaccines resulted in an Omicron focused response with lowest nAb and ACE2 binding inhibition results against ancestral and Delta compared with all groups primed with ancestral N1mΨ, irrespective of subsequent boost. This suggests that vaccines containing variants from the ancestral cluster should be assessed in immunological naïve individuals if variants of ancestral origin are actively circulating.

Our monovalent and bivalent CV0501 vaccines induced comparable nAb responses against Omicron subvariants, although only half the dose in the bivalent vaccine encodes BA.1 S protein. Our data demonstrated that boosting with bivalent CV0501/ancestral N1mΨ vaccine upon ancestral N1mΨ priming resulted in a tendency towards increased cross-neutralization of ancestral and Delta variants compared with boosting with CV0501 alone. Overall, our data point towards a potential benefit of bivalent vaccination for inducing a broad immune response. Similarly, previous mRNA vaccine studies in mice on an ancestral primed background demonstrated a broadening of immune responses upon boosting with ancestral/BA.1 bivalent vaccines compared with monovalent preparations [32,33].

Assessment of the monovalent CV0501 vaccine in a hamster challenge model demonstrated that CV0501 (at both 8 µg and 24 µg doses) protects against challenge with Omicron BA.2 by reducing weight loss and significantly decreasing viral load in the lungs to levels below or around the lower limit of detection compared with sham-immunized controls. Viral genome copies in the nose (representative of viral shedding) were significantly reduced but remained higher than levels observed in the lungs. In general, protection in the nose is more difficult to achieve than in the lower respiratory tract, which is more pronounced following heterologous compared with homologous virus challenge infection [34,35,36,37]. Differences in the BA.1 and BA.2 S protein sequence that affect nAb generation may have had an impact on nasal protection. BA.2 contains eight unique changes that are not present in BA.1. These mutations are located in the N-terminal domain (NTD) and the RBD, both known target regions for nAbs. In line with this, previous studies reported only partial protection against reinfection with BA.2 following previous infection with BA.1 in Syrian hamsters [38]. Overall, results in hamsters demonstrated the protective efficacy of CV0501 against BA.1 and BA.2 SARS-COV-2 in a widely accepted animal model for SARS-CoV-2 infection model [39].

## 5. Conclusions

Overall, vaccination with the SARS-CoV-2 BA.1 vaccine, CV0501, elicited robust dose dependent levels of nAbs against homologous BA.1 and other variants of SARS-CoV-2 in small animals. Importantly, CV0501 induced cross-nAbs against SARS-CoV-2 BA.5, a recommended vaccine component, and demonstrated potential as both a monovalent and bivalent vaccine candidate. CV0501 challenge studies in hamsters displayed a significant reduction of viral mRNA levels in the airways after challenge infection with BA.2, compared with the control. Based on these promising preclinical results, CV0501 has progressed into Phase 1 clinical trials in humans to assess suitability as a monovalent vaccine candidate (ClinicalTrials.gov Identifier: NCT05477186).

## Figures and Tables

**Figure 1 vaccines-11-00318-f001:**
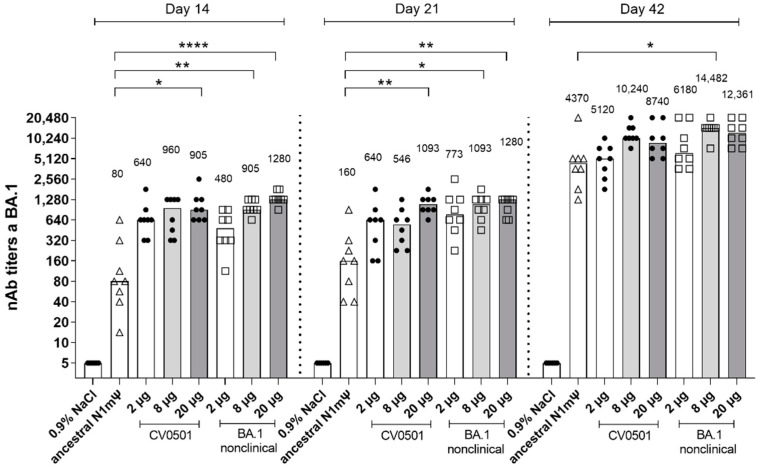
CV0501 and BA.1 nonclinical induce BA.1 specific viral nAb titers upon vaccination over time. Wistar rats (*n* = 8/group) were immunized i.m. on Days 0 and 21 with different doses of CV0501 (black circles) or comparator vaccine (ancestral N1mΨ [8 µg, triangles] or BA.1 nonclinical [squares]). nAb titers from rat serum were measured on 14-, 21- and 42-days post-immunization. Sera were analyzed after each timepoint and data combined into a single figure. Each symbol represents an individual animal and bars depict the median (values are indicated as number above each bar). Statistical analysis was performed on each timepoint separately. ANOVA and Kruskal-Wallis tests were used to compare all groups to ancestral N1mΨ and to 8 µg CV0501, respectively. Only statistically significant results are indicated. * *p* < 0.05, ** *p* < 0.01, **** *p* < 0.0001.

**Figure 2 vaccines-11-00318-f002:**
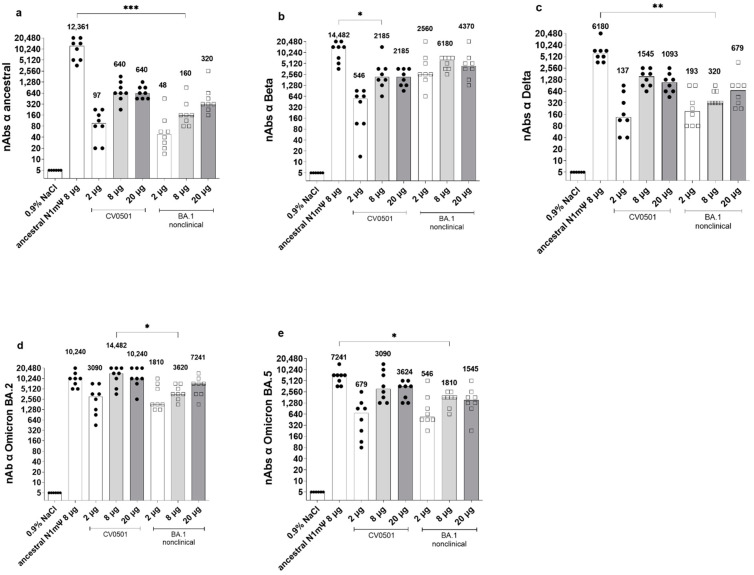
CV0501 and BA.1 nonclinical preferentially induce cross nAbs towards the Omicron subvariants. Wistar rats (*n* = 8/group) were immunized on Days 0 and 21 with 2, 8 or 20 µg CV0501 (circles) or BA.1 nonclinical (squares), 8 µg of ancestral N1mΨ (triangles), or 0.9% NaCl (control, *n* = 6). nAbs against the ancestral (**a**), Beta (**b**), Delta (**c**), BA.2 (**d**) and BA.5 (**e**) SARS-CoV-2 variants were assessed from serum taken on Day 42 post immunization. The symbols represent individual animals and the bars the median (median values are indicated as number above each bar). Statistical analyses were performed via pair-wise comparisons between ancestral N1mΨ, 8 µg CV0501 and 8 µg BA.1 nonclinical using ANOVA and Kruskal-Wallis test. Only significant results are indicated. * *p* < 0.05, ** *p* < 0.01, *** *p* < 0.001.

**Figure 3 vaccines-11-00318-f003:**
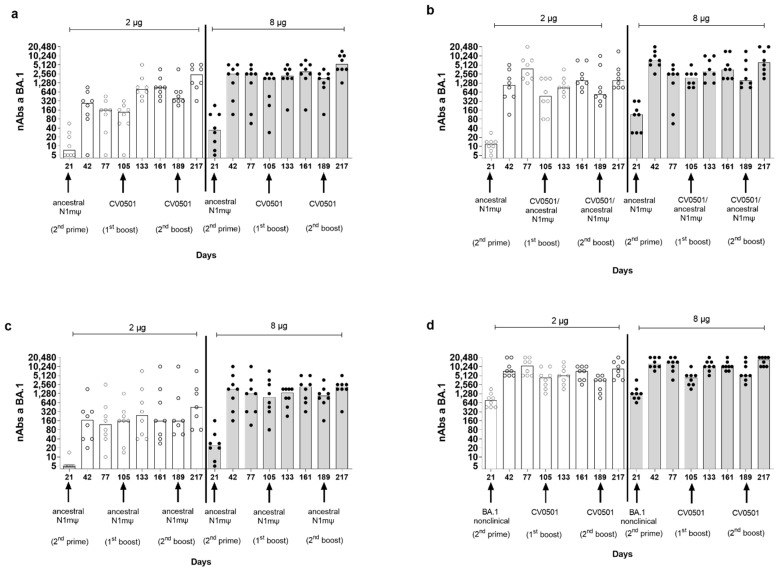
CV0501 and bivalent CV0501/ancestral N1mΨ boost nAb titers against BA.1 after priming with ancestral N1mΨ vaccine. Wistar rats (*n* = 8/group) were immunized i.m. (1st and 2nd prime) on Days 0 and 21 with either 2 µg (clear circles, white bars) or 8 µg (black circles, grey bars) doses of ancestral N1mΨ or BA.1 nonclinical. On Days 105 and 189, rats were given a third and fourth dose (1st and 2nd boost) of either (**a**,**d**) CV0501, (**b**) bivalent CV0501/ancestral N1mΨ (half doses of each), (**c**) ancestral N1mΨ (2 µg or 8 µg doses). nAbs against BA.1 SARS-CoV-2 were assessed in sera obtained on Days 21, 42, 77, 105, 133, 161, 189 and 217. Each symbol represents an individual rat and the bars represent the median value. Analyses of nAb titers were performed on separate days for each timepoint. One-way ANOVAs and Kruskal-Wallis tests were performed to compare nAb titers between Days 105 and 133 and between Days 189 and 217.

**Figure 4 vaccines-11-00318-f004:**
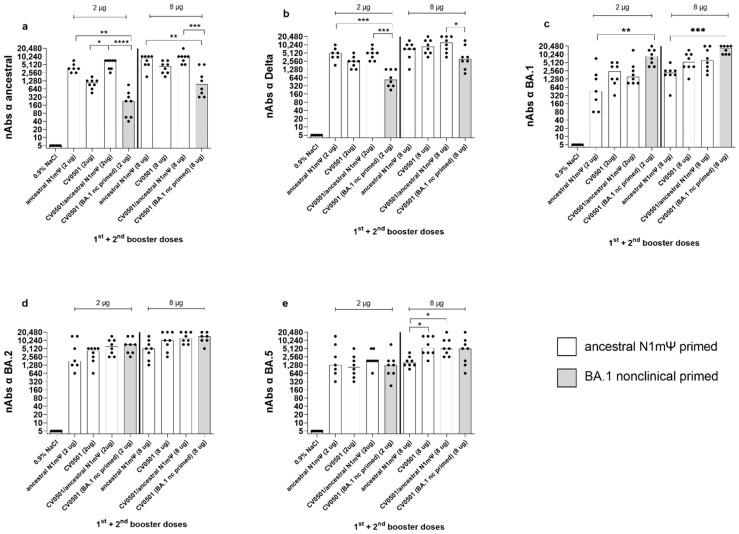
CV0501, ancestral N1mΨ and bivalent CV0501/ancestral N1mΨ boost cross-nAb titers after priming with ancestral N1mΨ vaccine. Wistar rats (*n* = 8/group) were immunized i.m. on Days 0 and 21 with 2 µg or 8 µg of ancestral N1mΨ (white bars) or BA.1 nonclinical (grey bars) and boosted on Days 105 and 189 with either CV0501, the bivalent combination CV0501/ancestral N1mΨ or ancestral N1mΨ. Animals immunized with 0.9% NaCl Buffer (*n* = 6/group) were used as negative controls. Neutralizing Abs against ancestral (**a**), Delta (**b**), BA.1 (**c**), BA.2 (**d**) or and BA.5 (**e**) SARS-CoV-2 variants were assessed in sera taken on Day 217. Each dot represents an individual animal and bars represents the median. One-way ANOVAs and Kruskal-Wallis tests for the 2 µg and 8µg doses were performed. * *p* < 0.05, ** *p* < 0.01, *** *p* < 0.001, **** *p* < 0.0001.

**Figure 5 vaccines-11-00318-f005:**
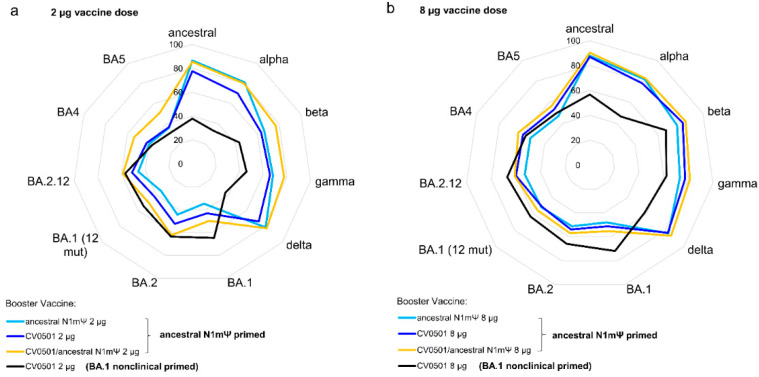
Priming of rats with ancestral N1mΨ results in induction of broader ACE2-binding inhibiting antibodies compared to priming with BA.1 nonclinical with little variance induced by selection of booster vaccine. Wistar rats (*n* = 8/group) were primed on Days 0 and 21 with 2 or 8 µg of ancestral N1mΨ or BA.1 nonclinical (black line) and boosted on Day 105 with either CV0501 alone (dark blue line and black line), ancestral N1mΨ alone (light blue line) or the bivalent combination CV0501/ancestral N1mΨ (orange line). Sera collected on Day 133 from rats receiving 2 µg vaccines was diluted 1:1600 (**a**) and from rats receiving 8 µg vaccines was diluted 1:3200 (**b**) for the ACE2 binding inhibition assay. Data are displayed as the percentage RBD-binding inhibition, with 0% being in the inner circle and 100% in the outmost circle.

**Figure 6 vaccines-11-00318-f006:**
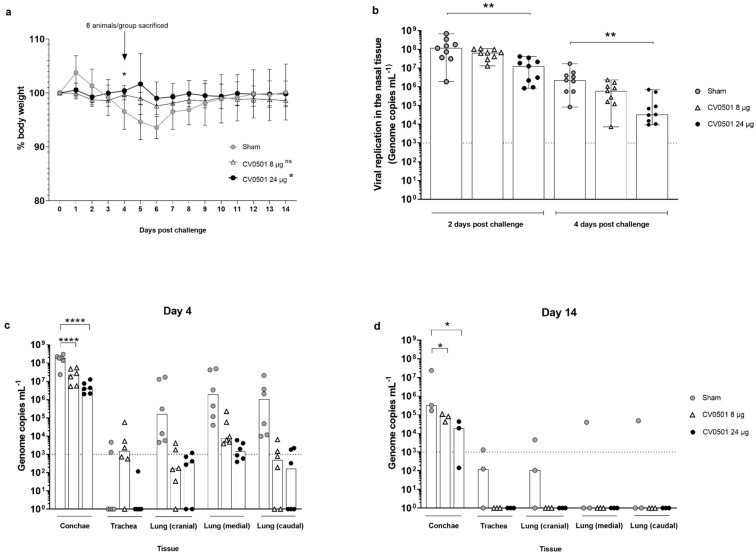
Immunization with CV0501 provided protection against Omicron BA.2 challenge in Syrian hamsters. Male Syrian hamsters (*n* = 9/group) received either 8 µg or 24 µg CV0501 or 0.9% NaCl (sham controls) on Days 0 and 28, followed by challenge with Omicron BA.2 (10^5^ TCID_50_/animal administered i.n. at 0.05 mL per nostril) on Day 56. (**a**) Percentage body weight was recorded daily between Days 0 and 14 post-challenge. Each dot represents the mean value of the group at the indicated time points. Error bars show the SD in each group. Statistical analyses were performed using Dunnett’s Test for multiple comparisons. (**b**) Viral replication in the nose was assessed in nasal wash samples collected on Days 2 and 4 post-challenge by RT-qPCR. Each dot represents an individual animal, bars represent the median value of the group at the various time points. Statistical analyses were performed for Days 2 and 4 days post-challenge separately using Kruskal-Wallis test and Dunnett‘s Test for multiple comparisons. The dashed line indicates the lower limit of detection for the assay. (**c**,**d**) Viral loads were assessed in conchae, trachea and lung (cranial, caudal and medial) samples collected on Day 4 (**c**) or on Day 14 (**d**) post-challenge. Each dot represents an individual animal, bars depict the median value of the group at the various time points. Statistical analysis was performed using two-way ANOVA testing. Samples exhibiting less than 10^3^ genome copies per mL (dashed line) were considered negative. * *p* < 0.05, ** *p* < 0.01, **** *p* < 0.0001.

**Figure 7 vaccines-11-00318-f007:**
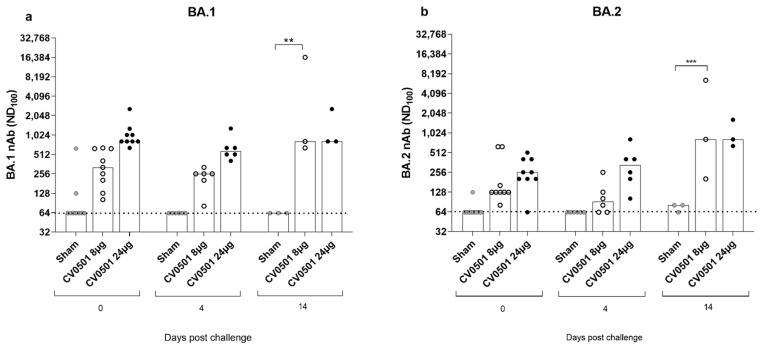
Significant induction of nAb against BA.1 and BA.2 Omicron variants observed after CV0501 vaccination. Hamsters (*n* = 9/group) were immunized on Days 0 and 28 with 8 µg or 24 µg doses of CV0501 (white and black circles respectively) or 0.9% NaCl (sham controls) (grey circles) and challenged on Day 56 with Omicron BA.2 (10^5^ TCID_50_/animal administered i.n. at 0.05 mL per nostril). nAbs against BA.1 (**a**) and BA.2 (**b**) were assessed in sera collected on Days 0 and 4 (6 animals/groups) or Day 14 (3 animals/groups) post-challenge. Virus nAb titers expressed as the neutralizing dose (ND) 100 of the sera. Each dot represents an individual animal, and the bars represent the median value of the group at the various time points. The horizontal dotted lines indicate the lower limit of detection of the assay. Statistical analysis was performed using two-way-ANOVA and Dunnett’s Test for multiple comparison. ** *p* < 0.01, *** *p* < 0.001.

**Table 1 vaccines-11-00318-t001:** Overview of mutations and mRNA-LNP vaccines used.

Name	Encoded SARS-CoV-2 Variant	mRNA Modification	AA Mutations in the Spike Protein * (Compared with Ancestral)	AA Mutations Specific to the RBD Region (Compared with Ancestral)
CV0501	BA.1 (12 RBD mutations)	N1mΨ	A67V, del69–70, T95I, G142D, del143–145, del211, L212I, ins214EPE, T547K, D614G, H655Y, N679K, P681H, N764K, D796Y, N856K, Q954H, N969K and L981F.	G339D, S371L, S373P, S375F, S477N, T478K, E484A, Q493R, G496S, Q498R, N501Y, Y505H
BA.1 nonclinical	BA.1 (15 RBD mutations)	N1mΨ	A67V, del69–70, T95I, G142D, del143–145, del211, L212I, ins214EPE, T547K, D614G, H655Y, N679K, P681H, N764K, D796Y, N856K, Q954H, N969K and L981F.	As above but with 3 additional mutations: K417N, N440K and G446S
Ancestral	Ancestral	N1mΨ	-	-

AA, amino acid; LNP, lipid nanoparticle; N1mΨ, N1-methylpseudouridine; mRNA, messenger ribonucleic acid; RBD, receptor binding domain; SARS-CoV-2, severe acute respiratory syndrome coronavirus-2. * Not including AA mutations in the RBD.

**Table 2 vaccines-11-00318-t002:** Analyses of median nAb production against SARS-CoV-2 variant BA.1 after booster vaccination with CV0501 alone, ancestral N1mΨ alone or bivalent CV0501/ancestral N1mΨ.

Group	1st Prime (Day 0) 2nd Prime (Day 21)	1st Boost (Day 105) 2nd Boost (Day 189)	Dose Concentration	Median nAb Titers								
				Day 21	Day 42	Fold-Increase Days 21–42	Day 105	Day 133	Fold-Increase Days 105–133	Day 189	Day 217	Fold-Increase Days 189–217
1	Ancestral N1mΨ	CV0501	2 µg	8	273	36	137	773	5.6	386	2450	6.3
			8 µg	34	2560	75	1810	2185	1.2	1810	5430	3.0
2	Ancestral N1mΨ	CV0501/ ancestral N1mΨ	2 µg	12	1093	91	480	905	1.9	546	1545	2.8
			8 µg	113	7241	64	1920	3090	1.6	1545	6180	4.0
3	Ancestral N1mΨ	Ancestral N1mΨ	2 µg	5	170	34	160	240	1.5	160	453	2.8
			8 µg	24	1810	75	960	1358	1.4	1092	2185	2.0
4	BA.1 nonclinical	CV0501	2 µg	773	7241	9	4370	5120	1.2	3620	8740	2.4
			8 µg	1280	12,361	10	4370	10,240	2.3	5120	17,481	3.4

N1mΨ, N1-methylpseudouridine; nAb, neutralizing antibody; SARS-CoV-2, severe acute respiratory syndrome coronavirus-2.

**Table 3 vaccines-11-00318-t003:** Analyses of median nAb production against SARS-CoV-2 variants after booster vaccination with CV0501 alone, ancestral N1mΨ alone, or bivalent CV0501/ancestral N1mΨ.

					Median nAb Titers vs. SARS-CoV-2 Variants on Day 217				
Group	1st Prime (Day 0) 2nd Prime (Day 21)	1st Boost (Day 105)	2nd Boost (Day 189)	Dose Concentration	Ancestral	Delta	BA.1	BA.2	BA.5
1	Ancestral N1mΨ	Ancestral N1mΨ	Ancestral N1mΨ	2 µg	3620	5120	452	1810	1280
				8 µg	8740	7241	2185	5120	1545
2	Ancestral N1mΨ	CV0501	CV0501	2 µg	1093	2560	2450	4370	1093
				8 µg	4370	8740	5431	10,240	5431
3	Ancestral N1mΨ	Bivalent CV0501/ancestral N1mΨ	Bivalent CV0501/ancestral N1mΨ	2 µg	7241	5120	1545	6180	1810
				8 µg	10,240	12,361	6180	12,361	5120
4	BA.1 nonclinical	CV0501	CV0501	2 µg	226	546	8740.4	7241	1280
				8 µg	960	3090	17,480.75	14,482	5120

N1mΨ, N1-methylpseudouridine; nAb, neutralizing antibody.

## Data Availability

The data presented in this study are available on request from the corresponding author. The data are not publicly available due to intellectual property rights.

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
