# Peer review of "Assessment of Immunogenicity and Efficacy of CV0501 mRNA-Based Omicron COVID-19 Vaccination in Small Animal Models"

_vaccines, 2023, doi:10.3390/vaccines11020318_

Round 1

Reviewer 1 Report

The manuscript brings a subject of great worldwide interest and focuses on developing new vaccines for COVID-19. The work is well presented, with adequate rationale, justification, results, and conclusions, a fact that deserves publication. However, before being accepted, the authors should answer the questions:

Minor points

-Oscillations in the results of some groups were detected. For example, Table 2, in which the results of group 1, day 105, were lower than those on day 21. The same can be observed with the concentration of 8 ug on day 133 than on day 42. Can this variation be due to the small number of animals used in each group?

-I suggest explaining in the manuscript how many times the experiments were repeated. Unfortunately, I could not find this information in the text.

-According to the rules, the state and city of the manufacturers must be informed

Author Response

Response to Reviewer 1 Comments

Comments and Suggestions for Authors

The manuscript brings a subject of great worldwide interest and focuses on developing new vaccines for COVID-19. The work is well presented, with adequate rationale, justification, results, and conclusions, a fact that deserves publication. However, before being accepted, the authors should answer the questions:

Thank you for your review and we appreciate the feedback provided. We have made amendments as suggested and these are highlighted in the manuscript. In addition, we have provided a response to each point below.

Point 1: Oscillations in the results of some groups were detected. For example, Table 2, in which the results of group 1, day 105, were lower than those on day 21. The same can be observed with the concentration of 8 ug on day 133 than on day 42. Can this variation be due to the small number of animals used in each group?

Response 1: While we did not observe lower values at day 105 compared with day 21 in Table 2, we do acknowledge post-boost viral nABs (day 133) to be lower than day 42 values, however, for all groups (except for Group 2 8 µg dose) these differences are still within the 2-fold variability of the VNT assay.

Although the increases in nAbs were not shown to be statistically significant (Figure 3), they do indicate that lower titres on day 21 could efficiently be boosted on day 42 after administration of a second priming vaccination. Booster vaccinations given on day 105 were able to increase nAb responses by day 133 but this increase was not significant. We hypothesize that one reason for the reduced dose response could be that the increased levels of circulating antibodies on day 105 could interfere with the recognition of the expressed spike protein by memory B cells necessary for an efficient recall response. Since we have not presented any experimental data to support this hypothesis, we have not added this to the manuscript.

Point 2: I suggest explaining in the manuscript how many times the experiments were repeated. Unfortunately, I could not find this information in the text.

Response 2:  Each animal study was performed once, in line with the principles of 3Rs for animal testing from Russell and Burch. This information has been added to the manuscript (page 3, line 116). Although the setting is not identical, the study in previously primed animals confirms that CV0501 can induce a BA.1 specific immune response in naïve animals.

Point 3: According to the rules, the state and city of the manufacturers must be informed.

Response 3:  For suppliers in the USA, we have given city and state but for suppliers in other countries, we have given city and country. All changes have been implemented in the manuscript and highlighted. We have also listed the details below:

  • Male Syrian hamsters aged 11 weeks were purchased from Janvier Labs (Le Genest-Saint-Isle, France) (line 112)

  • Z-clot activator tubes (Sarstedt AG, Nümbrecht, Germany) -line 126

  • SARS-CoV-2 peptide library (JPT, PM-SARS2-SMUT08-1, Berlin, Germany) -line 153

  • ELISpot Rat interferon-gamma (IFNγ) (Cat: EL585 by R&D Systems, Minneapolis, MN). – line 155

  • MagPlex beads (Luminex, Austin, TX,) – line 160

  • (AMG Activation Kit for Multiplex Microspheres, Anteo Technologies, Brisbane, QLD, Australia [#A-LMPAKMM-400]) – line 161

  • Streptavidin-Phycoerythrin (PE) (Dunn Labortechnik GmbH, Asbach, Germany)– line 171

  • FLEXMAP3D instrument (Luminex, Austin, TX) – line 174

  • TissueLyser II (Qiagen, Hilden, Germany). – line 198

  • NucleoMag Vet kit (Macherey Nagel, Dueren, Germany – line 203

Reviewer 2 Report

The manuscript evaluated immunogenicity and efficacy of CV0501 mRNA-based COVID-19 vaccination in in vivo study with rats and hamsters. This study focused on titration of Abs against SARS-CoV-2, especially, Omicron BA spike proteins. In addition, the protective efficacy of CV0501 against BA.2 infection was evaluated in hamsters.

The work is laborious and informative. The manuscript is basically well written although a little complicated. I have some comments regarding descriptions in the manuscript.

The formats of Table 1 and Table 2 are broken. They should be corrected.

Figure 3. Is the statistical description (page 11, lines 377-379) in the figure caption necessary? No statistical description is shown in Figure 3.

Figure 4. Page 12, line 20. “(median values are indicated as number above each bar)” No numbers are shown in Figure 4.

Figure 5. The figure caption described that “Priming of rats with ancestral was compared to priming with BA.1 nonclinical”. Where are the results of priming with BA.1 nonclinical? The difference between figure a and b should be more clearly shown.

Figure 6. The differences of a, b, c, and d should be more clearly shown.

Figure 7. The difference between a and b should be more clearly shown.

I cannot find captions of supplemental figures.

Figure S3. The difference between a and b should be more clearly shown.

Minor comments

Page 7, lines 274,277. “IFN-g” should be “IFN-gamma (Greek letter)”.

Page 9, line 314. “numerical higher” should be “numerically higher”.

Page 9, line 351. Add “against” before “ancestral”.

Page 14, line 444. “although…” should be “although only reductions observed in the conchae were significantly lower”.

Author Response

Response to Reviewer 2 Comments

Comments and Suggestions for Authors

The manuscript evaluated immunogenicity and efficacy of CV0501 mRNA-based COVID-19 vaccination in in vivo study with rats and hamsters. This study focused on titration of Abs against SARS-CoV-2, especially, Omicron BA spike proteins. In addition, the protective efficacy of CV0501 against BA.2 infection was evaluated in hamsters.

The work is laborious and informative. The manuscript is basically well written although a little complicated. I have some comments regarding descriptions in the manuscript.

Thank you for your thorough review and we appreciate the feedback provided. We have made amendments as suggested and these are highlighted in the manuscript. In addition, we have provided a response to each point below.

Point 1: The formats of Table 1 and Table 2 are broken. They should be corrected.

Response 1: We have changed the horizontal lines in Table 1 to match those in Table 2 and 3.

Point 2: Figure 3. Is the statistical description (page 11, lines 377-379) in the figure caption necessary? No statistical description is shown in Figure 3.

Response 2: We felt it was important to inform the reader that we had performed statistical analysis on the data in this figure, which is why we added the details to the figure caption. We have also discussed that there were no statistically significant increases in nAb titers for this figure in the text (Page 9, line 330) and because of this, we thought it would be important to state which statistical tests were performed.

Point 3: Figure 4. Page 12, line 20. “(median values are indicated as number above each bar)” No numbers are shown in Figure 4.

Response 3: Thank you for bringing this to our attention. This has now been removed from figure 4 legend (page 12 line, 388).

Point 4: Figure 5. The figure caption described that “Priming of rats with ancestral was compared to priming with BA.1 nonclinical”. Where are the results of priming with BA.1 nonclinical? The difference between figure a and b should be more clearly shown.

Response 4: We have made some edits to figure 5 in line with point 4.

  1. Figure (a) shows the data for 2 µg doses of vaccines, and (b) for 8 µg doses. This labelling has now been added to the figure to make the differences clearer for the reader.
  2. The figure legend has been edited and the term immunized replaced by primed (Page 13, line 395). The rats primed with BA.1 nonclinical are represented by the black line and for further clarity, we have added labelling to show this group was primed with BA.1 nonclinical vaccine.

Point 5: Figure 6. The differences of a, b, c, and d should be more clearly shown.

Response 5: We have changed the y axis title of Fig 6b to include ‘viral replication in the nasal tissue’. For figures c and d we have added Day 4 and Day 14 as titles to the figures to show the differences between these figures. 

Point 6: Figure 7. The difference between a and b should be more clearly shown.

Response 6: Both BA.1 and BA.2 titles have been added to Figure 7 to highlight the differences between a and b. 

Point 7: I cannot find captions of supplemental figures.

Response 7: Apologies as it seems that you were not send the PDF/Word document with the supplementary data and captions for the supplementary figures. We will upload the document with the captions included again.

Point 8: Figure S3. The difference between a and b should be more clearly shown.

Response 8: Figure S3 a) shows the data for 2 µg vaccine doses and b) shows the data from 8 µg vaccine dose. These details have been added as titles to the figure to highlight the differences.

Minor comments

Point 9: Page 7, lines 274,277. “IFN-g” should be “IFN-gamma (Greek letter)”.

Response 9: Thank you for highlighting this and it has now been amended to IFN-γ-producing T cells.

Point 10: Page 9, line 314. “numerical higher” should be “numerically higher”.

Response 10: Thank you for highlighting this and it has now been amended to numerically higher

Point 11: Page 9, line 351. Add “against” before “ancestral”.

Response 11: Against has now been added to this line.

Point 12: Page 14, line 444. “although…” should be “although only reductions observed in the conchae were significantly lower”.

Response 12: Amended as requested.